**Data Availability Statement:** All relevant data are within the manuscript and its Supporting Information files.

# The impact of assisted reproductive technology on prenatally diagnosed fetal growth restriction in dichorionic twin pregnancies

Viola Seravalli [1]*, Lorenzo Maoloni[1], Lucia Pasquini[2], Sara Bolzonella[3], Giovanni Sisti[4], Felice Petraglia[3], Mariarosaria Di Tommaso[1]

1 Department of Health Sciences, University of Florence, Florence, Italy, 2 Department of Obstetrics & Gynecology, Fetal Medicine Unit, Careggi University Hospital, Florence, Italy, 3 Department of Clinical and Experimental Biomedical Sciences, University of Florence, Florence, Italy, 4 Department of Obstetrics and Gynecology, Lincoln Medical and Mental Health Center, Bronx, NY, United States of America

☯ These authors contributed equally to this work.
* viola.seravalli@unifi.it

## Abstract

### Objective

Whether the use of assisted reproductive technologies (ART) affects the outcome of twin pregnancies is still a matter of debate. Previous studies have evaluated the association between birth weight and ART, without a clear distinction between fetal growth restriction (FGR), a condition at higher risk of adverse outcome, and constitutionally small for gestational age (SGA) fetuses.

The aim of this study was to determine whether dichorionic (DC) twin pregnancies obtained by ART have a greater risk of developing FGR, defined by accurate ultrasound criteria, than those spontaneously conceived (SC), and to compare the severity of ultrasound features in the growth restricted fetuses.

### Methods

A retrospective study was conducted on DC twin pregnancies delivered between 2010 to 2018 at a tertiary hospital. Twin pregnancies conceived spontaneously were compared with those obtained via in vitro fertilization (IVF) or intracytoplasmic sperm injection (ICSI), after exclusion of cases with major fetal or uterine malformations. The primary outcome was the incidence of FGR. Secondary outcome was the rate of SGA neonates, defined by a birth weight less than the 10th percentile. The ultrasound characteristics of the growth restricted fetuses in the two groups were also compared. The groups were compared using univariate and multivariate analyses.

### Results

Six hundred and seventy-eight DC twin pregnancies were identified. Of these, 367 (54.1%) conceived via IVF/ICSI and 311 (45.9%) conceived spontaneously. The incidence of FGR

**Funding:** The authors received no specific funding for this work.

**Competing interests:** The authors have declared that no competing interests exist.

was not significantly different between the ART and the SC groups (7.9% vs 8.4% respectively, p = 0.76, adjusted OR 0.84, 95% CI 0.53–1.32). Growth restricted fetuses of the two groups showed similar occurrence of an estimated fetal weight less than the 3$^{rd}$ percentile, similar abnormalities in Doppler studies and similar gestational age at diagnosis. There was no difference in the incidence of delivery of an SGA neonate (p = 0.47) or in the rate of maternal complications and preterm delivery between the groups.

## Conclusions

Twin pregnancies conceived by assisted reproductive technologies do not have a higher risk of ultrasound-diagnosed FGR than spontaneously conceived twin pregnancies, and fetuses diagnosed with growth restriction in the two groups show similar severity of the ultrasound findings.

## Introduction

In singleton pregnancies, the use of assisted reproductive technology (ART) techniques has been consistently associated with adverse maternal and perinatal outcomes, such as preterm delivery, hypertensive disorders, reduced fetal growth and perinatal mortality [1–4]. On the other hand, studies comparing ART with spontaneously conceived (SC) twin pregnancies have yielded conflicting results. Lack of stratification for maternal age and parity, as well as differences in study population may explain the inconsistent findings. Some studies do not specify the technique used for assisted reproduction, or the chorionicity of the pregnancies studied. Monochorionic (MC) twin pregnancies are known to have a higher risk of adverse fetal outcome compared to dichorionic (DC) twins [5], and to occur more rarely in ART pregnancies compared to those conceived spontaneously.

With respect to the association between mode of conception and reduced fetal growth in twins, birth weight was often the only measure used to define cases with insufficient fetal growth [6–8] and, even in the few studies where prenatal estimation of fetal weight (EFW) was considered, fetal growth restriction was defined by an EFW below a given threshold, with no evaluation of fetal doppler [9, 10]. Therefore, no distinction was made between cases of fetal growth restriction (FGR), at higher risk of adverse outcome, and constitutionally small for gestational age (SGA) fetuses, which may have similar outcomes to their normally grown counterparts. Recently, the key diagnostic features for FGR in twins have been determined by expert consensus, using a Delphi procedure [11], in order to reduce the heterogenicity in the definition of this condition and to increase the ability to compare and combine findings of future studies on this topic. These features include a combination of low EFW and abnormal fetal Doppler studies [11].

The aim of this study was to determine whether DC twin pregnancies conceived by ART have a greater risk of developing FGR, defined by appropriate ultrasound criteria, than those conceived spontaneously, and to compare the severity of ultrasound features of the growth restricted fetuses.

## Methods

A retrospective study was conducted on DC twin pregnancies delivered between 2010 to 2018 at Careggi University Hospital, a tertiary hospital, in Florence, Italy. The study was approved

by the Institutional Ethics Committee (*Comitato Etico Regionale per la Sperimentazione Clinica della Regione Toscana)*. Given the retrospective nature of the study, the ethics committee, complying with our Institution's guidelines for clinical observational and retrospective studies, waived the requirement for informed consent when it was impracticable to contact the patient to obtain the consent. In all patients an ultrasound was performed in the first trimester to confirm chorionicity and gestational age. Pregnancy dating was based on last menstrual period in spontaneously conceived twins and confirmed by measurement of fetal crown–rump length of the larger twin [12]. In ART-conceived twins, dating was based on the date of egg retrieval in fresh cycles and on embryonic age in frozen-thawed cycles. Exclusion criteria were conception via intrauterine insemination or ovulation induction, evidence of major fetal malformations, uterine malformations or congenital infections, and delivery before 22 weeks.

Pregnancies conceived spontaneously were compared to those obtained via in vitro fertilization (IVF) or intracytoplasmic sperm injection (ICSI). The primary outcome was the incidence of FGR, defined according to the ultrasound criteria determined by expert consensus in DC pregnancies [11]. These are based on ultrasound fetal biometry and fetal Doppler studies, as shown in Table 1. Ultrasound Doppler studies included assessment of the umbilical artery (UA), middle cerebral artery (MCA) and ductus venosus (DV) blood flow. Fetal growth restriction was defined as early- or late-onset depending on the gestational age at diagnosis (before or after 32 weeks). All cases of FGR diagnosed antenatally were evaluated by ultrasound at the Center for Fetal Medicine of Careggi hospital, the same institution at which all patients enrolled in this study delivered. The protocol for ultrasound monitoring of pregnancies complicated by FGR at our Center include growth scans performed every 2 weeks, and Doppler studies performed at least weekly, or more often depending on the severity of Doppler flow abnormality, until the threshold for delivery is reached. The ultrasound characteristics of the growth restricted fetuses in the SC- and the ART- groups were also compared. As fetuses affected by FGR had multiple scans performed throughout pregnancy, in order to compare the ultrasound features between ART and SC twins we used the worst Doppler findings recorded for each case.

Secondary outcome was the incidence of delivery of an SGA neonate, defined by a birth weight less than the 10th percentile according to Italian reference curves (INeS reference chart) [13]. The group of SC twin pregnancies was compared with the group of ART-conceived pregnancies (IVF or ICSI) using the chi-square test or Fisher's exact test for categorical variables, and the student t-test or Mann-Whitney test for continuous variables. Normality of continuous data was tested using the Shapiro–Wilk test. Multiple logistic regression analysis was used to calculate the odds ratios (OR) and 95% confidence intervals (CI) for the independent associations of IVF/ICSI with each outcome after adjustment for potential confounding

**Table 1. Ultrasound criteria for the definition of FGR in dichorionic twin pregnancies (adapted from Khalil et al, 2019 [11]).**

| Dichorionic twin pregnancies | |
| --- | --- |
| **SOLITARY** | Parameter that alone is sufficient for diagnosis |
| Estimated fetal weight (EFW) of one of the twins <3rd centile | |
| **CONTRIBUTORY** | 2 out of 3 contributory parameters are required irrespective of which parameter |
| EFW <10th centile of one of the twins | |
| EFW discordance ≥25% | |
| Umbilical artery (UA) pulsatility index (PI) of the smaller twin >95th centile | |

variables such as maternal age and parity. Statistical analysis was performed using SPSS version 24.0 and a p-value of <0.05 was considered significant.

## Results

Six hundred and seventy-eight DC twin pregnancies met the inclusion criteria. Of these, 367 (54.1%) were conceived via IVF or ICSI and 311 (45.9%) were conceived spontaneously. Maternal and pregnancy characteristics are reported in Table 2. Gestational age at delivery was similar between groups. Women in the ART group were older and more frequently nulliparous. Pregnancy complications such as gestational diabetes and hypertensive disorders were observed more frequently in the ART group compared to the SC group at univariate analysis (p = 0.03 and 0.007, respectively). After multiple regression analysis adjusting for maternal age and parity, ART conception was not independently associated with neither of these outcomes (aOR 1.08, 95%CI 0.67–1.74, p = 0.75 for gestational diabetes and aOR 1.28, 95%CI 0.58–2.81, p = 0.55 for hypertensive disorders). The incidence of preeclampsia was also similar between groups. Sixty-one percent (n = 416) of women delivered preterm (<37 weeks), and 31% of these preterm births followed spontaneous onset of labor. There was no significant difference in the rate of preterm delivery between the SC and ART group for any of the gestational age ranges considered (Table 2).

Fetal and neonatal outcomes are reported in Table 3. Overall, 96 pregnancies (14.2%) were complicated by FGR of at least one fetus, and in 5 of these cases FGR was associated with

**Table 2. Maternal characteristics and pregnancy outcome according to mode of conception.**

| Characteristic | All pregnancies (n = 678) | Spontaneous conception (n = 311) | ART conception (n = 367) | p |
|---|---|---|---|---|
| Maternal age (years) | 35 (32, 39) | 34 (30, 37) | 37 (34, 42) | <0.001 |
| Advanced maternal age (>35 y) | 395 (58.3%) | 133 (42.8%) | 262 (71.4%) | <0.001 |
| Body mass index (kg/m$^2$)* | 21.8 (20.2, 24.5) | 21.9 (20.3, 24.8) | 21.6 (20.1, 24.3) | 0.17 |
| Nulliparous | 484 (71.4%) | 161 (51.8%) | 323 (88%) | <0.001 |
| Mode of ART | | | | |
| • IVF | 342 (49.7%) | | | |
| • ICSI | 25 (3.6%) | | | |
| Hypertensive disorders | 45 (6.6%) | 12 (3.9%) | 33 (9%) | 0.55[†] |
| Preeclampsia | 18 (2.7%) | 6 (1.9%) | 12 (3.3%) | 0.28 |
| Gestational diabetes | 127 (18.7%) | 47 (15.1%) | 80 (21.8%) | 0.75[†] |
| Intrahepatic cholestasis | 56 (8.3%) | 26 (8.4%) | 30 (8.2%) | 0.93 |
| GA at delivery | 36 (34, 37) | 36 (34, 37) | 36 (34, 37) | 0.38 |
| Term birth (≥37 weeks) | 262 (38.6%) | 130 (41.8%) | 132 (36%) | 0.12 |
| Late preterm (34$^{+0}$–36$^{+6}$ w) | 266 (39.2%) | 119 (38.3%) | 147 (40.1%) | 0.63 |
| Moderate preterm (32$^{+0}$–33$^{+6}$w) | 68 (10%) | 25 (8%) | 43 (11.7%) | 0.11 |
| Low preterm (28$^{+0}$-31$^{+6}$w) | 51 (7.5%) | 22 (7.1%) | 29 (7.9%) | 0.68 |
| Very-low preterm (<28w) | 31 (4.6%) | 15 (4.8%) | 16 (4.4%) | 0.77 |
| Cesarean section | 571 (84.1%) | 247 (79.4%) | 324(88.3%) | 0.002 |
| Elective c-section | 337 (59%) | 143 (46.0%) | 194 (52.9%) | 0.07 |
| Emergency c-section | 234 (41%) | 104 (33.4%) | 130 (35.4%) | 0.59 |

ART, assisted reproductive technology; IVF, in vitro fertilization; ICSI intracytoplasmatic sperm injection, GA, gestational age

Data are presented as n (%) or median and interquartile range (IQR, 25[th], 75[th] percentile)

*Missing data in 20 patients

[†] Adjusted for maternal age and parity

**Table 3. Fetal and neonatal outcomes in the study population.**

| Outcome | All fetuses (n = 1356) | Spontaneous conception (n = 622) | ART conception (n = 734) | p | Adjusted odds ratio (95%CI) * |
|---|---|---|---|---|---|
| FGR | 110 (8.1%) | 52 (8.4%) | 58 (7.9%) | 0.76 | 0.84 (0.53–1.32) |
| IUFD | 6 (0.4%) | 4 (0.6%) | 2 (0.3%) | 0.42 | |
| Birth weight (g) | 2350 (1907, 2650) | 2390 (1915, 2720) | 2290 (1900,2600) | 0.24$^{\dagger}$ | |
| SGA | | | | | |
| • <10$^{th}$ perc. | 273 (20.1%) | 130 (21.1%) | 143 (19.6%) | 0.47 | 0.86 (0.63–1.19) |
| • <5$^{th}$ perc. | 149 (11.0%) | 74 (12.1%) | 75 (10.3%) | 0.30 | 0.78 (0.52–1.17) |
| 5-min Apgar score <7 | 18 (1.4%) | 9 (1.5%) | 9 (1.2%) | 0.72 | |
| Arterial pH<7.20 $^{\ddagger}$ | 73 (5.4%) | 29 (6.6%) | 44 (7.8%) | 0.48 | |

ART: assisted reproductive technology; FGR fetal growth restriction; IUFD, intrauterine fetal demise, SGA, small for gestational age.

Data are presented as n (%) or median and interquartile range (IQR, 25$^{th}$, 75$^{th}$ percentile)

*adjusted for maternal age and parity

$^{\dagger}$ adjusted for gestational age at delivery and parity

‡ Data available for 906 neonates

preeclampsia. Since 14 pregnancies had FGR of both fetuses, the total number of fetuses with an ultrasound diagnosis of FGR was 110 (8.1% of all fetuses). The incidence of FGR was not significantly different between the IVF/ICSI group and the SC group (7.9% vs 8.4% respectively, p = 0.76, aOR 0.84, 95%CI 0.53–1.32). The ultrasound characteristics and frequency of Doppler abnormalities of the 110 fetuses with diagnosis of FGR are reported in Table 4. Sixty per cent of cases of FGR had early onset (<32 weeks), and the most frequent Doppler abnormality observed was an increase in umbilical artery (UA) pulsatility index (PI). Fetal demise of the FGR fetus occurred in three cases. Of these, two had absent end-diastolic velocity (AEDV), and one had reversed EDV (REDV) in the UA. All three cases had very early onset (<26 weeks) FGR, oligohydramnios and an EFW <3$^{rd}$ percentile. When the ultrasound

**Table 4. Ultrasound characteristics of FGR cases (n = 110).**

| Variable* | n (%) (tot = 110) | Spontaneous conception (n = 52) | ART conception (n = 58) | p |
|---|---|---|---|---|
| GA at diagnosis (weeks) | 31.3 (25.4, 33.3) | 31.8 (25.8, 33.3) | 31.1 (25.3, 33.5) | 0.62 |
| Early onset FGR (<32 weeks) | 66 (60%) | 31 (59.6%) | 35 (60.3%) | 0.94 |
| EFW <3$^{rd}$ percentile | 56 (50.9%) | 27 (52.9%) | 29 (50%) | 0.76 |
| Weight discordance between twin 1 and twin 2 (%) | 21.8 (12.0,28.6) | 18.3 (12.0, 27.4) | 22.5 (11.7, 28.9) | 0.71 |
| Abnormal UA Doppler* | 93 (84.5%) | 46 (90.2%) | 47 (81%) | 0.18 |
| UA PI >95$^{th}$ percentile | 75 (68.2%) | 39 (76.5%) | 36 (62.1%) | 0.10 |
| AEDV | 10 (9.1%) | 3 (5.9%) | 7 (12.1%) | 0.33 |
| REDV | 8 (7.3%) | 4 (7.8%) | 4 (6.9%) | 1.00 |
| Brain sparing (MCA PI <5$^{th}$ percentile) | 40 (36.3%) | 18 (35.3%) | 22 (37.9%) | 0.78 |
| Increased DV PI | 6 (5.5%) | 3 (5.9%) | 3 (5.2%) | 1.0 |
| Reversed DV a-wave | 0 | 0 | 0 | - |
| Oligohydramnios (MVP<2 cm) | 11 (10%) | 7 (13.5%) | 4 (6.9%) | 0.25 |
| Fetal demise | 3 (2%) | 2 (3.8%) | 1 (1.7%) | 0.46 |

*When repeated ultrasounds were performed, the worst Doppler finding was used.

Data are presented as n (%) or median and interquartile range (IQR, 25$^{th}$, 75$^{th}$ percentile)

FGR, fetal growth restriction; GA, gestational age; EFW, estimated fetal weight

UA, umbilical artery; PI, pulsatility index; AEDV, absent end-diastolic velocity; REDV, reversed end-diastolic velocity; MCA, middle cerebral artery; DV, ductus venosus; MVP, maximum vertical pocket.

characteristics of the growth-restricted fetuses in the SC and ART-conception groups were compared, similar incidence of abnormal Doppler studies of the UA, MCA and DV, similar incidence of an EFW less than the 3$^{rd}$ percentile, and similar gestational age at diagnosis were observed (Table 4). Oligohydramnios and fetal demise also occurred with similar frequency.

The incidence of delivery of an SGA neonate with birth weight less than the 10$^{th}$ or less than the 5$^{th}$ percentile was also similar between groups (p = 0.47 and 0.30, respectively) (Table 3). After adjustment for maternal age and parity, the results were confirmed (aOR 0.86, 95%CI 0.63–1.19 for SGA <10$^{th}$ percentile and aOR 0.78, 95% CI 0.52–1.17 for SGA <5$^{th}$ percentile). The incidence of intrauterine fetal demise was low and did not differ between groups (0.6 ad 0.3%, p = 0.42).

## Discussion

Our study showed that DC twin pregnancies conceived by IVF/ICSI do not have a higher risk of FGR, defined by consensus-based diagnostic criteria, than spontaneously conceived twins, regardless of maternal age and parity, and that the ultrasound features of fetuses diagnosed with growth restriction are similar between the two modes of conception. To our knowledge, no previous study has analyzed the difference in prenatally diagnosed FGR between twin pregnancies conceived spontaneously and pregnancy conceived via ART. In fact, most studies investigating the effect of different modes of conception on fetal growth and perinatal outcome used birth weight as the main outcome measure [6–8], or, in some cases, used an ultrasound-EFW below a given threshold to define fetal growth restriction, with no mention of the results of Doppler studies [9, 10]. Ultrasound biometry is the gold standard for assessment of fetal size, and Doppler velocimetry of multiple fetal vascular territories (both arterial and venous) is fundamental to identify the growth-restricted fetus at greatest risk for neonatal morbidity and mortality, and to distinguish it from a "constitutionally" small fetus [14, 15]. Doppler interrogation of fetal vessels is also widely accepted as part of the management of pregnancies complicated by FGR, to predict fetal hypoxemia and acidemia and to determine the appropriate timing for delivery. This applies to both singleton and twin pregnancies [11, 14, 16]. Therefore, studies aiming to investigate factors that may be associated with the risk of developing FGR cannot ignore the use of fetal Doppler studies for the diagnosis of such disorder. Inconsistences in the diagnostic criteria for FGR employed in existing studies hinder the ability to compare or combine their findings. For this reason, consensus definitions of FGR in singletons [16] and twins [11] have been published recently, and the criteria agreed by experts should be used in future studies on FGR.

Two previous studies on DC twins used the EFW to diagnose fetal growth restriction: one used the 10$^{th}$ percentile for gestational age as cut off, and the other the 3$^{rd}$ percentile [9, 10]. Neither of them found a difference in the incidence of reduced fetal size, nor in the incidence of low and very low birth weight between the ART and SC groups [9, 10]. A recent large met-analysis of studies comparing outcomes in pregnancy conceived naturally or through ART [17] reported an overall relative risk of intrauterine growth restriction of 1.08 (95%CI 0.88–1.34, p = 0.45) from the combined analysis of three studies, two of which were the aforementioned studies, that defined fetal growth restriction based on prenatal EFW, while the third one used a birth weight less than the 10$^{th}$ percentile as the outcome variable [6].

Our finding of a similar incidence of delivery of an SGA neonate between twin pregnancies conceived via IVF/ICSI and pregnancies conceived spontaneously is consistent with findings of previous studies [1, 2, 7, 8, 18]. Maternal outcomes such as hypertensive disorders, gestational diabetes and intrahepatic cholestasis in our study were similar between IVF/ICSI and SC pregnancies, after adjusting for maternal age and parity. Although these were not primary

outcomes of the present study, it is noteworthy that these results are in accordance with data reported in several studies [6–10, 19–21]. We did not observe any significant difference in the rate of preterm delivery between the SC- and IVF/ICSI group for any of the gestational age ranges considered. This finding is consistent with previous observational studies [6, 8, 9, 18, 19, 21], but not with a recent metanalysis where the combined analysis of twelve studies showed an increased risk of preterm birth and very preterm birth in ART twin pregnancies compared to those conceived spontaneously [17].

Currently there is a general belief that ART pregnancies are at higher risk of adverse outcome compared to those conceived spontaneously. This could be in part related to the fact that ARTs are associated with higher rate of multiple pregnancies, which themselves carry increased risk of adverse outcome. However, when the analysis is restricted to singleton pregnancies, several studies, including metanalyses, suggest that ART-pregnancies have a significantly worse maternal and perinatal outcome than non-assisted singletons [2–4, 22, 23]. Which aspect of assisted reproduction technology poses most risk and how this risk can be minimized has not been determined [22]. Furthermore, the association between ART and adverse outcome described for singletons has not been demonstrated for twin pregnancies, as many inconsistent findings from studies published in the last two decades have emerged. The large metanalysis by Qin et al. of fifteen cohort studies comprehensively assessed obstetric outcomes of DC twin pregnancies after ART, and suggested that rate of adverse outcomes such as placenta previa, preterm birth, low birth weight and congenital malformations were indeed higher in ART-twin pregnancies compared to SC pregnancies, while no difference was observed for maternal hypertension, gestational diabetes, perinatal mortality, placental abruption, premature rupture of membranes, postpartum hemorrhage and delivery of an SGA neonate [17]. The authors also recognized that their results have to be viewed with caution because of the substantial heterogeneity observed among studies.

The reason why the use of ART seems to be associated with a higher rate of adverse outcomes in singletons, but not as much in twin pregnancies, could be that the well known increase in the risk of maternal and perinatal complications associated with carrying a multiple pregnancy masks any possible effect of the mode of conception on these outcomes. Finally, it should be underlined that the majority of studies on the relationship between mode of conception and adverse outcome do not evaluate parental factors related to subfertility, and it is therefore difficult to separate the contribution of infertility per se from that of the ART treatment [23]. This remains a major challenge that future studies should try to confront.

Strengths of our study include the use of previously published ultrasound criteria, determined by expert consensus [11], to accurately define FGR in our cohort. Another strength of this study is that it used a relatively large and homogeneous group of patients: we only included DC twins, to avoid the effect of monochorionicity itself on the rate of adverse outcome, and all cases of FGR diagnosed were evaluated at the center for Fetal Medicine of Careggi hospital, the same institution at which all patient enrolled in this study delivered. This ensured higher homogeneity in the ultrasound assessment of cases of FGR, and also in the management of the enrolled patients, as the same obstetric protocol was used to manage all twin pregnancies, limiting the risk of bias. In addition, among the ART techniques we only considered IVF/ICSI, excluding from our analysis ovulation induction alone or in combination with intrauterine insemination, which, despite being infertility treatments, in our opinion cannot be grouped with IVF/ICSI when investigating the possible effect of ART on pregnancy outcomes, because of the broad difference between these techniques, nor should be considered analogous to spontaneous conception.

The main limitation of this study is the retrospective nature of data collection. However, in all cases of FGR, all data on fetal biometry and Doppler studies necessary to evaluate our main

outcome were available, as they are part of the routine evaluation of growth restricted fetuses at our Fetal Medicine Center, and all images and results are stored in the ultrasound software used at our Institution, so they are easily accessible and verifiable retrospectively. Additional limitations include the lack of maternal data on smoking history and on the cause of infertility in the ART group.

In conclusion, our findings indicate that twin pregnancies conceived through IVF/ICSI do not show an increase in the incidence of FGR or SGA compared to spontaneously conceived pregnancies, and that fetuses diagnosed with growth restriction in the two groups show similar ultrasound features. Maternal and pregnancy outcomes also appear to be similar. These results are useful for counselling patients carrying ART-conceived twins. The main clinical implication of our results is that the protocol for ultrasound surveillance of uncomplicated dichorionic twins should continue to be the same irrespective of mode of conception.

## Supporting information

**S1 Table. Logistic regression analysis using gestational diabetes as the dependent variable.**
(DOCX)

**S2 Table. Logistic regression analysis using hypertensive disorders as the dependent variable.**
(DOCX)

**S3 Table. Logistic regression analysis using fetal growth restriction as the dependent variable.**
(DOCX)

**S4 Table. Logistic regression analysis using delivery of an SGA neonate with birthweight $<10^{th}$ percentile as the dependent variable.**
(DOCX)

**S5 Table. Logistic regression analysis using delivery of an SGA with birthweight $<5^{h}$ percentile as the dependent variable.**
(DOCX)

**S1 Full Dataset.**
(SAV)

## Author Contributions

**Conceptualization:** Viola Seravalli, Lorenzo Maoloni, Lucia Pasquini, Giovanni Sisti, Felice Petraglia, Mariarosaria Di Tommaso.

**Data curation:** Viola Seravalli, Lorenzo Maoloni, Sara Bolzonella.

**Formal analysis:** Viola Seravalli, Giovanni Sisti.

**Methodology:** Viola Seravalli, Lucia Pasquini, Felice Petraglia, Mariarosaria Di Tommaso.

**Supervision:** Viola Seravalli, Felice Petraglia, Mariarosaria Di Tommaso.

**Validation:** Felice Petraglia, Mariarosaria Di Tommaso.

**Writing – original draft:** Viola Seravalli.

**Writing – review & editing:** Lorenzo Maoloni, Lucia Pasquini, Sara Bolzonella, Giovanni Sisti, Felice Petraglia, Mariarosaria Di Tommaso.

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
