## [Decision Letter · Decision Letter 0]

20 Feb 2020

PONE-D-19-31923

The impact of assisted reproductive technology on prenatally diagnosed fetal growth restriction in dichorionic twin pregnancies

PLOS ONE

Dear Authors,

Thank you for submitting your manuscript to PLOS ONE. After careful consideration, we feel that it has merit but does not fully meet PLOS ONE’s publication criteria as it currently stands. Therefore, we invite you to submit a revised version of the manuscript that addresses the points raised during the review process.

We would appreciate receiving your revised manuscript by 31st March. To enhance the reproducibility of your results, we recommend that if applicable you deposit your laboratory protocols in protocols.io, where a protocol can be assigned its own identifier (DOI) such that it can be cited independently in the future. For instructions see: http://journals.plos.org/plosone/s/submission-guidelines#loc-laboratory-protocols

We look forward to receiving your revised manuscript.

Kind regards,

Salvatore Andrea Mastrolia, M.D.

Academic Editor

PLOS ONE

Journal Requirements:

2. Thank you for including your ethics statement in the manuscript: 'The study was approved by the Institutional Ethics Committee.'

3. In ethics statement in the manuscript and in the online submission form, please provide additional information about the patient records/samples used in your retrospective study. Specifically, please ensure that you have discussed whether all data/samples were fully anonymized before you accessed them and/or whether the IRB or ethics committee waived the requirement for informed consent. If patients provided informed written consent to have data/samples from their medical records used in research, please include this information.

Reviewers' comments:

Reviewer's Responses to Questions

**Comments to the Author**

1. Is the manuscript technically sound, and do the data support the conclusions?

Reviewer #1: Yes

2. Has the statistical analysis been performed appropriately and rigorously? 

Reviewer #1: Yes

3. Have the authors made all data underlying the findings in their manuscript fully available?

Reviewer #1: Yes

4. Is the manuscript presented in an intelligible fashion and written in standard English?

Reviewer #1: Yes

5. Review Comments to the Author

Reviewer #1: The study have been conducted rigorously with appropriate replication and the statistical analysis has well performed.

The strength of the study is the importance of the topic, the limitation is the retrospective approach.

The authors indicate that hypertensive disorders were observed more frequently in ART group however they don’t describe preeclampsia cases especially in FGR and SGA group.

More details about cesarean section category could improve the fetal outcome severity.

6. PLOS authors have the option to publish the peer review history of their article (what does this mean?). If published, this will include your full peer review and any attached files.

Reviewer #1: No

---

## [Author Response · Author response to Decision Letter 0]

28 Feb 2020

Thank you for giving us the chance to further improve our manuscript. Below are all questions raised, followed by our response. We have now modified the manuscript according to the reviewers’ comments

We hope that these modifications would allow the paper to be accepted in PLOS ONE

Reviewer #1: The study have been conducted rigorously with appropriate replication and the statistical analysis has well performed.

The strength of the study is the importance of the topic, the limitation is the retrospective approach.

The authors indicate that hypertensive disorders were observed more frequently in ART group however they don’t describe preeclampsia cases especially in FGR and SGA group.

R. We have now included data on the incidence of preeclampsia in the overall cohort, and in the SC and ART-group (see table 2). In addition , we have added a comment on cases of IUGR associated with preeclampsia in the result section (line 135)

More details about cesarean section category could improve the fetal outcome severity:

R. Of the 571 cesarean section performed, 59% (337/571) were elective c-sections, while the remaining 41% were emergency c-section, performed either during labor or for maternal or fetal indications in non-laboring patients. While the overall incidence of c-section was significantly higher in the IVF/ICSI group compared to the SC group, no significant difference was found in the incidence of these subcategories (elective or emergency c-section) between groups. All these results have now been included in table 2.

---

## [Decision Letter · Decision Letter 1]

16 Mar 2020

The impact of assisted reproductive technology on prenatally diagnosed fetal growth restriction in dichorionic twin pregnancies

PONE-D-19-31923R1

Dear Authors,

We are pleased to inform you that your manuscript has been judged scientifically suitable for publication and will be formally accepted for publication once it complies with all outstanding technical requirements.

With kind regards,

Salvatore Andrea Mastrolia, M.D.

Academic Editor

PLOS ONE

Reviewers' comments:

Reviewer's Responses to Questions

**Comments to the Author**

1. If the authors have adequately addressed your comments raised in a previous round of review and you feel that this manuscript is now acceptable for publication, you may indicate that here to bypass the “Comments to the Author” section, enter your conflict of interest statement in the “Confidential to Editor” section, and submit your "Accept" recommendation.

Reviewer #1: All comments have been addressed

2. Is the manuscript technically sound, and do the data support the conclusions?

Reviewer #1: Yes

3. Has the statistical analysis been performed appropriately and rigorously? 

Reviewer #1: Yes

4. Have the authors made all data underlying the findings in their manuscript fully available?

Reviewer #1: Yes

5. Is the manuscript presented in an intelligible fashion and written in standard English?

Reviewer #1: Yes

6. Review Comments to the Author

Reviewer #1: (No Response)

7. PLOS authors have the option to publish the peer review history of their article (what does this mean?). If published, this will include your full peer review and any attached files.

Reviewer #1: No

---

## [Editor Report · Acceptance letter]

3 Apr 2020

PONE-D-19-31923R1 

The impact of assisted reproductive technology on prenatally diagnosed fetal growth restriction in dichorionic twin pregnancies 

Dear Dr. Seravalli:

I am pleased to inform you that your manuscript has been deemed suitable for publication in PLOS ONE. Congratulations! Your manuscript is now with our production department. 

With kind regards,

on behalf of

Dr. Salvatore Andrea Mastrolia 

Academic Editor

PLOS ONE